# Performance Test and Thermal Insulation Effect Analysis of Basalt-Fiber Concrete

**DOI:** 10.3390/ma15228236

**Published:** 2022-11-21

**Authors:** Xiao Zhang, Shuo Zhang, Song Xin

**Affiliations:** 1College of Safety and Environmental Engineering, Shandong University of Science and Technology, Qingdao 266590, China; 2State Key Laboratory of Mining Disaster Prevention and Control Co-Found by Shandong Province and the Ministry of Science and Technology, Shandong University of Science and Technology, Qingdao 266590, China; 3College of Transportation, Shandong University of Science and Technology, Qingdao 266590, China

**Keywords:** basalt fiber, high geothermal roadway, compressive strength, SEM images, impermeability, thermal insulation, numerical simulation, cost-benefit analysis

## Abstract

This paper examines the feasibility of applying inorganic thermal-insulating concrete in high geothermal roadways in underground coal mines. This innovative material is based on a mixture of ceramsite, glazed hollow beads, cement, and natural sand, enhanced with varying degrees of basalt fibers. Fibers were used as a partial substitute in the mixture, in the following volumes: 0% (reference specimen), 5%, 10%, 15%, and 20%. Their compressive strength, permeability resistance, and thermal conductivity were studied. A high content of fibers tends to entangle into clumps during mixing, resulting in a significant reduction in the mechanical properties of compressive strength. The appropriate amount of fiber content can improve impermeability, and the permeability height of 5% fiber concrete was reduced by 22.5%. Experiments on thermal behavior showed that an increase of basalt fibers leads to a significant reduction in thermal conductivity. For concrete containing 20% fiber, the thermal conductivity for the reference specimen (0%) in the wet state was reduced from 0.385 W/(m∙°C) to 0.098 W/(m∙°C). There was a slight increase in thermal conductivity when the temperature increased from 30 °C to 60 °C. Despite the reduced mechanical strength, the resulting concrete is well-suited for use in the insulation of underground roadways, as numerical simulations showed that insulating concrete with optimal fiber content (15%) can reduce the average temperature of the wind flow in a high ground temperature roadway of 100 m in length in a mine by 0.3 °C. The final cost-benefit analysis showed that insulating concrete has more economic benefits and broad development prospects when applied to high geothermal roadway cooling projects.

## 1. Introduction

China is rich in coal resources. With the gradual development of the national economy and changes in society, the industrialization process is accelerating, and the demand for coal resources is increasing. Relevant research studies have shown that many coal mines have carried out high-intensity mining for a long time to meet the rapid economic development, resulting in fewer and fewer coal resources in shallow areas near the surface. With these changes, together with improvements in the mechanization level of mining, coal mines have gradually progressed to deep mining [1].

When mining deep resources, the problem of thermal damage caused by high geothermal heat is often encountered. The effects of high temperatures on the roadway within a mine, such as a deteriorating construction environment, a reduction in labor productivity, threats to the health and safety of workers [2,3], and a reduction in the safety, stability, and durability of the mining structure [4], cannot be ignored. Coal mines use mechanical cooling measures to counteract heat damage [5]. However, such cooling methods rely on the configuration of suitable large-scale refrigeration equipment, which significantly increases the construction capital of coal mines; the equipment maintenance and electricity costs are prohibitive. In the problematic context of the mining of coal resources and decreasing economic efficiency, a new model for heat-damage management with low costs and high efficiency is urgently needed.

Thermal insulation materials are widely developed in various contemporary construction industries because of their low costs, good insulation effects, and easy methods of preparation [6,7,8]. In the case of mining, thermal insulation materials provide a new concept in managing heat damage in mines: i.e., spraying thermal insulation materials on the surfaces of mine roadways to inhibit the heat exchange between the wind flow and the surrounding rock. There are precedents for applying organic insulation materials (such as polyethylene foam and polyurethane foam) in many deep mines in South Africa [9]. However, organic insulation materials have significant drawbacks in the unique underground environment; notably, in the event of a fire, they will burn rapidly and release toxic and hazardous gases, threatening the lives of coal miners. Moreover, these new polymer materials are relatively expensive and not cost-effective for large-scale use in underground roadways. The unique environment of underground coal mines is different from general above-ground building environments. Additional factors need to be considered when choosing a suitable insulation material for use in an underground coal mine, such as fire resistance, anti-static requirements, water resistance, and the elimination of harmful gas emissions, all of which are related to the safety of underground workers.

Ceramsite is made, at high temperatures, of muddy shale, clay, gangue, and fly ash. It is characterized by low density, low water absorption, high closed porosity, high strength, and internal pores filled with gas, which can be used as a barrier for heat transfer and for good thermal insulation [10]. Adding a glazed hollow bead to concrete can improve a structure’s seismic performance and durability, fire resistance, and thermal insulation [11]. After strict selection measures, we chose ceramsite and glazed hollow beads as the base materials for downhole insulating concrete.

Basalt fiber is a pollution-free “green industrial material” for the 21st century. It has excellent mechanical and thermal properties, stable chemical properties, and compatibility [12]. Adding basalt fibers to concrete can inhibit the generation and development of early microcracks and macroscopically strengthen the integrity of the matrix [13]. Basalt fibers improve the permeability, carbonation, acid and alkali resistance, and frost resistance of concrete, extending its service life in extreme environments [13,14,15]. Mehran Khand’s incorporation of basalt fibers in concrete resulted in substantial improvements in the stress–strain response, peak stress, modulus of elasticity, peak strain, ultimate staining, toughness, and specific toughness at elevated temperatures [16]. 

Basalt is a natural inorganic fiber that has no harmful effects on humans or the environment during its production and application. It is abundant in most parts of the world and is cheap to mine and process. Basalt does not require additives in the production process, which provides it with an additional advantage in terms of cost. In addition, basalt fibers have very low thermal conductivity (0.032 W/(m∙°C)). Therefore, basalt-fiber thermal-insulated concrete is an ideal candidate for supporting high geothermal roadways in mines, with excellent potential for improving building insulation and reducing the cooling costs of coal mines. Most researchers studying the mechanical properties of basalt-fiber concrete use very low percentages of fiber reinforcement (1% to 5%), and the compressive strength, splitting tensile strength, and flexural strength of concrete can be significantly improved [17,18,19]. It is necessary to investigate the incorporation of higher percentages of basalt fibers in concrete to achieve the desired thermal return.

Currently, numerical simulations are widely used to study the heat transfer between rock and wind flow in mining roadways. Zhou et al. [20] developed a finite-difference numerical model to calculate the frozen length near the tunnel entrance, which agreed with the measured length. Kang [21] established a two-dimensional axisymmetric model of coupled convective heat transfer based on the k-ε turbulence equation and studied the temperature field of high-temperature tunnel airflow; an empirical formula for the effective ventilation distance was established. Previous studies made significant progress in wind-flow-perimeter rock-heat transfer. As field measurements are expensive and time-constrained, numerical simulations of the heat transfer from the tunnel envelope are very mature; therefore, we used numerical simulations to investigate the thermal insulation effect of basalt-fiber concrete.

We focused on the potential use of basalt fibers as reinforcing additives in thermal-insulating concrete. Basalt fibers were used as partial substitutes in concrete mixtures at different volume levels: 0% (reference specimen), 5%, 10%, 15%, and 20%. The average length of the fibers that were used was approximately 10 mm. An experimental test procedure was used to study the properties of the hardened specimens, including their apparent density, compressive strength, permeability, and thermal conductivity in the dry and wet states. Then, Fluent numerical-simulation software was applied to analyze the cooling effect after spraying insulating concrete on the surface of a high-temperature mine tunnel in Shandong, China. Finally, the cost-effectiveness of the thermal-insulating concrete was analyzed. The innovative material reduced energy consumption in mines by slowing down rock exotherms, within the broader context of low carbon and environmental protection. At the same time, the simulations provided an essential guide for selecting cooling measures for high geothermal mines.

## 2. Materials and Experimental Tests

### 2.1. Materials

We used 42.5 ordinary Portland cement with a density of 3.10 g/cm^3^; the compressive strengths of 3 d and 28 d are 28.9 MPa and 52.3 MPa, respectively. We used river sand of 2.78 fineness modulus, with density of 2.63 g/cm^3^, Llevel II, and bulk density of 1.50 g/cm^3^. The ceramsite particle size was ~0.8 mm to 1.5 mm with loose bulk density of 600 kg/m^3^, numerical tube pressure of 3.5 MPa, water absorption of 7%, thermal conductivity of 0.039 W/(m·°C), and porosity of 25%. The particle sizes of the glazed hollow beads were ~0.5 mm to 1.0 mm, with bulk density of 80 kg/m^3^, thermal conductivity of 0.027 W/(m·°C), a closed-cell rate of 96%, and water absorption of 8%. The basalt fiber was 10 mm in length, with a diameter of 15 μm, density of 2.65 g/cm^3^, tensile strength of ~3300 MPa to 4500 MPa, and ultimate elongation of ~2.4% to 3.0% [22]. The experimental water was drinking water. The physical properties of ceramsite, glazed hollow beads, and basalt fiber are shown in Table 1, Table 2 and Table 3, respectively. The appearances of ceramsite, glazed hollow beads, and basalt fiber are shown in Figure 1.

The core of the preparation of thermal-insulation concrete is basalt fiber, as an additive to improve the performance of shotcrete. Therefore, the ratio-design method should consider not only the ratio of ordinary shotcrete, but also the compatibility of basalt fiber with concrete. Due to the high ground stress, high ground temperature, and high-karst water pressure in deep mines, the concrete shot on a roadway’s surface needs to have strong mechanical strength, thermal-insulation properties, and impermeability. These factors ultimately make the ratio of insulated concrete different from that of ordinary shotcrete. The proportion of concrete that is suitable for high geothermal roadways was selected, especially to meet the thermal-insulation requirements [23,24]. Combined with the ratio-design specification and experience, the reference ratio of thermal-insulating concrete was derived after several trials and adjustments. The mass ratio of cement, sand, and ceramsite was 1.2:1.89:1.02; the water–cement ratio was 0.55; and the glazed hollow beads accounted for 10% of the volume of the mixture. Basalt fibers were used as a partial replacement for the mixed mortar in the control specimens. The fiber-replacement levels, by volume, were 0% (reference mortar), 5%, 10%, 15%, and 20%.

Mixing is a very complex process in the preparation of thermal-insulation concrete. The purpose of mixing is not only to achieve a homogeneous mixing of the materials, but also to consider the homogeneous dispersion of the fibers in the cement matrix. Before making the mortar, the ceramsite and the glazed hollow beads were soaked in water for 3 h to achieve better bonding with the cement. Then, the dry fiber, cement, and sand were mixed thoroughly in a standardized mortar mixer. The moistened ceramsite, glazed hollow beads, and water were gradually added to the mixer until a homogeneous mixture was obtained. The concrete was loaded into the mold and fully compacted on a vibrating table. After the specimens were stripped, they were placed in a curing chamber at 20 ± 2 °C and 98% relative humidity. The process of specimen fabrication is shown in Figure 2. To measure hardening properties, samples of cubes, cone platforms, and cuboids were prepared in sizes (150 × mm 150 mm × 150 mm), (175 mm × 185 mm × 150 mm), and (150 mm × 150 mm × 50 mm), respectively. Cube and cone platform specimens measure apparent density, compressive strength, and impermeability after 28 d curing. Rectangular specimens were used to measure thermal conductivity. The thermal conductivity was measured in wet and dry conditions, when the conservation specimens were maintained for 3 d. The specimens were dried in a drying oven at 50 ± 2 °C for 5 h. The low drying temperature prevented the samples from developing microcracks. The thermal conductivity of the specimens was measured in a moist state at temperatures of 30 °C, 40 °C, 50 °C, and 60 °C.

### 2.2. Experimental Testing

The performance tests of the hardened specimens included the apparent dry density, determined by geometric measurements and weighing. According to “Standard for test method of mechanical properties on ordinary concrete”, the compressive strength of the specimens was measured using an AGX-250 electronic universal testing machine. This tester has a maximum load capacity of 2000 kN and a constant rate of 1 mm/min [25]. The hydrostatic pressure method was used to measure the permeability resistance of the insulating concrete. The impermeability test was carried out according to the “long-term performance and durability of PC method” [26], using a concrete permeability meter of type SS-1.5. We started with 0.1 MPa, held at 0.1 MPa for 2 h, increased to 0.3 MPa, then increased by 0.1 MPa every 8 h. When water seepage occurred on the surface of three of the six specimens, the test was stopped, and the water pressure at that time was recorded. In this study, if the water pressure was increased to 4.0 MPa and the specimen did not seep after 1 h, the sample was split, and the permeation height was measured. Based on the steady-state plate method, the thermal conductivity of the specimens was measured using the DRPL-I thermal conductivity tester, according to “Thermal insulation-Determination of steady-state thermal resistance and related properties-Guarded hot plate apparatus” [27]. The principle of this method is to maintain a stable temperature on the hot surface, heat transfer through the specimen to the cold surface, measure the heat flow transferred, then calculate the thermal conductivity and thermal resistance based on the thickness and heat transfer area of the specimen, and measure the thermal conductivity of the specimen with the DRPL test system. The instrument is shown in Figure 3.

## 3. Experimental Results and Discussion

### 3.1. Apparent Density of Dry-Insulating Concrete

Figure 4 shows the variation of apparent density versus fiber volume for dry concrete. The dry density decreased with the increase of fiber volume. The value ranged from 1607 kg/m^3^ for concrete without fibers to 1304 kg/m^3^ for concrete with 20% fibers, a reduction of approximately 18% in dry density. In order to have a fresh cement mortar with good compatibility, the fiber concrete required more water than the reference specimen, resulting in a highly porous matrix during the drying process. In addition, many tiny pores were created around the fibers, reducing the insulating concrete’s density [28]. The first advantage of using basalt fibers for insulating concrete is that it makes the specimen lighter.

### 3.2. Compressive Strength

In order to resist the strain on the surrounding rock, a completed roadway of an underground coal mine development needs to be shotcrete in full section, especially a roadway dug in the coal seam. It is also necessary to install an anchor net before the shotcrete. The concrete in a roadway is a compressed element and its strength is essential to ensure the safe use of the roadway. The compressive strength of shotcrete can only be put to use after it meets the standard requirements. Therefore, the effect of fiber incorporation on the compressive strength of concrete is worth exploring. Concrete specimens were prepared with four different volume contents of fibers (5%, 10%, 15%, 20%) and compared with concrete without fibers (0%) to study the effect of fiber content on concrete strength.

Three sets of concrete specimens of each fiber content were prepared and tested for compressive strength after 28 days of curing, and the average compressive strength values are shown in Figure 5. The minimum value of compressive strength (20 MPa) allowed in the “Standard and Safety Technical Measures for Underground Coal Mine Roadway Grouting” [29] is also marked in Figure 5. It can be seen from Figure 5 that the compressive strength of concrete decreased with the increase in fiber content, and the addition of fiber reduced the compressive strength from 36.7 MPa to 17.4 MPa. The compressive strengths were lower than those of the reference specimen (0%). The compressive strength of concrete with 5% fiber content was closest to the reference specimen, and the strength decreased by 10.1%. The strength of concrete with 20% fiber content decreased by 52.5% and could not even meet the minimum standard of compressive strength. The addition of fiber had a negative influence on the compressive strength of concrete. 

According to Asim et al. [30], the fibers make the concrete more porous, which may be a limiting factor for the mechanical properties. In order to significantly reduce the thermal conductivity, we added a large number of basalt fibers to the concrete. However, when too much fiber is used, it twists into clumps during the mixing process and causes local strength reduction. Under the pressure load, cracks first appeared in the concrete matrix and gradually formed penetrating cracks (Figure 6), leading to the failure of the specimen. The agglomerated fibers in the matrix were the main factors that led to the reduction in compressive strength.

### 3.3. Impermeability

When the mining depth is more than 800 m, the karst water pressure will be greater than 3 MPa. At the same time, the native fissures in the surrounding rock of the roadway gradually develop under the influence of mining coal, forming seepage channels that result in high-temperature fissure water entering the extraction space and more severe heat damage in the mine. The seepage resistance of concrete is closely related to the underground thermal environment, so seepage resistance is an essential index for evaluating thermal-insulation concrete. In this paper, the hydrostatic pressure method was used to analyze the seepage resistance of concrete.

Two sets of concrete specimens with four fiber volume contents were prepared, along with two sets of reference specimens (0%), which were tested for permeability after 28 days of curing. The average values of permeability heights are shown in Figure 7. Figure 7 shows that the permeability height of the insulating concrete does not decrease with the increase of fiber content. Adding 5% and 10% fibers to the reference specimens reduced the permeability height of the concrete by 22.5% and 4.2%, respectively. In contrast, the permeability height of the concrete increased by 9.2% and 20.8% with the addition of 15% and 20% fibers, respectively. The results showed that the best permeability resistance was achieved at 5% fiber content. When the fiber volume content was more than 15%, there was an evident agglomeration phenomenon, which provided a channel for water infiltration and led to a severe decrease in impermeability.

### 3.4. SEM Microscopic Images

Figure 8 shows the SEM images of the cross-section of the reference specimen. It can be observed that the sand grains, ceramsite, and glazed hollow beads were well encapsulated by the cement, giving the matrix compactness and the absence of cracks. The disrupted glass beads can be observed in the local magnified view. The walls of the glass beads were tightly bonded to other materials, which explains their high mechanical strength and hardness.

Figure 9 shows the cross-section of the concrete sample containing 5% fiber. The raw material of basalt fiber is basalt, mainly composed of SiO_2_ and Al_2_O_3_. The SiO_2_ content reaches 45% to 60% and has a composition similar to that of silicate cement. Although the surface of basalt fiber is smooth, it has a natural compatibility with cement [31]. As seen from Figure 9, a portion of the fiber was firmly embedded in the matrix, and a certain amount of energy was consumed by the pulling action of the fiber when subjected to stress. In the local magnification, it can be observed that the fiber surface was covered with a thick layer of cementitious material, and also mixed with tiny spherical particles, indicating that there was a good bond between the fiber and the cement matrix. In addition, the fiber was not easily pulled out under the action of external force, which macroscopically improved the mechanical properties of concrete, as well as the durability performance. The small amount of fiber had better dispersion during the mixing process, bonding tightly with the concrete matrix and reducing the pores and holes in the matrix.

Figure 10 shows the cross-section of the concrete containing 20% fiber. When the fiber volume content was too high, some fibers were not mixed uniformly. They tended to be entangled together, and we observed the formation of basalt-fiber masses in Figure 10. There was little cement content inside the fiber mass, and there was no bonding between the fibers, forming many pores. The agglomerated fibers also attracted moisture from the surrounding area, lowering the water content of other materials, which was not conducive to curing the substrate. The fiber clusters negatively affected the compactness of the concrete and became weak points of strength.

### 3.5. Thermal Conductivity

Our research aimed to prepare shotcrete materials with good thermal-insulation properties to effectively slow down the heat dissipation from the surrounding rock. Thermal conductivity is a physical quantity that reflects the ability of a material to transfer heat, and the thermal-insulation properties of shotcrete are quantified using thermal conductivity. Underground coal mines are dark and humid; the relative humidity of the air is above 80% all year round, and the high ground temperature of the surrounding rock is generally ~30 °C to 60 °C. The tests were conducted in conjunction with actual downhole conditions. The thermal conductivity was measured in the wet state on the third day of curing, and in the dehydrated state. The thermal conductivity was also measured at temperatures of 30 °C, 40 °C, 50 °C, and 60 °C when the specimen was wet. Three sets of concrete specimens were prepared for each fiber content. The average of the three values was taken as the thermal conductivity for that fiber content.

Figure 11 shows the thermal conductivity of insulating concrete (in dry and wet conditions) at 30 °C, as a function of fiber content. It can be seen that the thermal conductivity decreases with increasing fiber content. In the wet state, it decreases from 0.385 W/(m∙°C) for 0% fiber concrete to 0.098 W/(m∙°C) for 20% fiber concrete. As a result, the addition of 20% fibers improved the thermal insulation of concrete by approximately 74.5%. The thermal conductivity of basalt fiber is 0.038 W/(m∙°C), which is much smaller than that of other materials in concrete, and the fiber mixed with other materials reduces the overall thermal conductivity. Moreover, the increase in porosity in fiber concrete is also a factor in the reduction of thermal conductivity. 

The thermal conductivity of concrete with different fiber contents was generally smaller in the dry state than in the wet state, with the most enormous difference in the reference specimen, with a 24.6% difference in thermal-insulation properties. This can be explained by the high-water content of wet concrete. At normal temperature and normal pressure, the thermal conductivity of liquid water is approximately 22 times higher than that of air. The dry air in the pores is replaced by moist air and water, and the diffusion of water vapor and the movement of water molecules play a major role in heat transfer [32]. Therefore, the greater the water content of concrete, the greater the thermal conductivity. To accurately calculate the thermal load, it is of great significance to consider the thermal conductivity of the humid-state insulation concrete.

The relationship between the thermal conductivity and the temperature of the concrete specimens in the wet condition is shown in Figure 12. It can be seen that the thermal conductivity of all specimens showed an approximately linear increasing trend with the increase of temperature. An increase in temperature from 30 °C to 60 °C increased the thermal conductivity of the control sample and the 20% fiber-insulated concrete by approximately 21% and 13%, respectively. The increase in temperature intensifies the molecular thermal motion and enhances the three basic modes of heat transfer [33]. The increase in temperature enhances the heat transfer from the concrete solid and the heat convection from the liquid in the pores. In addition, the thermal radiation between the pore walls is enhanced with increasing temperature. However, heat radiation transfers very little heat and can be neglected [34]. Therefore, the thermal conductivity of most materials increases with increasing temperature [35].

### 3.6. Optimal Fiber Content Selection Based on the Correlation Matrix Method

The correlation matrix method (RMA) is a comprehensive system evaluation method that is often used in system-engineering systems, which mainly use matrices to represent the quantitative relationships among evaluation indicators. The key to the correlation-matrix-system evaluation method is to use the pair-by-pair comparison method to determine each evaluation index’s weights and scales and to obtain the final system evaluation value by weighting. We used the correlation-matrix method to comprehensively evaluate concrete with different fiber contents and select the best basalt-fiber content.

#### 3.6.1. Determination of Evaluation Indicators and Their Weights

We determined a scheme for adding four kinds of fiber content to concrete: 5%, 10%, 15%, and 20%. The four evaluation indicators were apparent density, compressive strength, impermeability, and thermal insulation. The results of the evaluation are shown in Table 4. The evaluation indicators were compared using the pair-by-pair comparison method to obtain the weights of each indicator in the evaluation system; the comparison results are shown in Table 5. As shown in Table 5, the weights of the four indicators, apparent density, compressive strength, impermeability, and thermal insulation performance, were 0, 1/3, 1/6, and 1/2, respectively. The obtained weights showed that the compressive strength and thermal insulation performance had a more significant proportion in the system evaluation and were key evaluation indicators.

#### 3.6.2. Calculation of Comprehensive Evaluation Value

According to the weights of the system-evaluation indices, combined with the evaluation scale and the scoring results, the comprehensive evaluation values of different fiber contents were obtained using the calculation method of the correlation matrix, which as follows.
(1)$$V_{i}=\overset{n}{\underset{i=1}{\sum }}\overset{m}{\underset{j=1}{\sum }}\alpha_{j}\cdot V_{ij}$$
where *V_i_* is the comprehensive evaluation value, *n* is the number of evaluation programs, $\alpha_{j}$ represents the evaluation indicators, $V_{ij}$ is the evaluation index scale, and *m* is the number of evaluation indicators.

According to the established systematic-evaluation system, *n* = 4 and *m* = 4 in the calculation; the correlation matrix was constructed as shown in Table 6. The results of the overall evaluation value calculated using the correlation-matrix method showed the best overall performance of the insulating concrete with 15% fiber content.

## 4. Numerical Simulation of the Thermal Insulation Effect of Sprayed Thermal-Insulating Concrete on the Roadway Surface

The ANSYS Fluent numerical simulation software was used to simulate the spraying of plain concrete and insulating concrete on the roadway surface. Previous studies generally treated the surrounding rock as zero thickness and set the rock wall temperature as a constant, using either temperature-boundary conditions or convective-heat-transfer-coefficient-boundary conditions [21,36]. However, as the ventilation time increased, heat transfer within the rock and convective heat transfer between the rock wall and the airflow were coupled [37]. Considering the interaction of heat transfer in solids and convection in fluids, solid structures were added to the model to significantly improve the accuracy of the simulation. In this study, the coupled heat-transfer model of the roadway was established by considering the thickness of the surrounding rock. The cooling effect of thermal-insulation materials was studied by comparing and analyzing the temperature field, the wall temperature, and the degree of convective heat transfer on the roadway surface.

### 4.1. Geometric Modeling and Meshing

In this study, a section of a high-temperature mine in Shandong Province, China, at a depth of 1150 m below ground with a concentrated flat roadway, was used as the object of the study. The roadway was located in the rock at the bottom of the coal seam and had a service life of ~5 to 7 years. The CFD method was used to study the heat transfer between the surrounding rock–air flow. Specific geometric parameters were measured practically, and a full-size 3D geometric model of the three-centered archway was established by applying Rhino 6.0 software, as shown in Figure 13a. The roadway was 100 m long, 4.9 m high, and 4.6 m wide, with a concrete support layer of 0.2 m and a surrounding rock thickness of 30 m. The model consisted of the air channel, the concrete support layer, and the surrounding rock. Figure 13b shows the model’s cross-sectional profile.

The quality of the delineated mesh determined the accuracy of the simulation solution results. The model was meshed using Fluent’s Meshing software, and contained 6,339,966 elements and 1,396,927 nodes. Skewness and orthogonality are the two principal quality measures in ANSYS meshing. Skewness determines how close a face or cell is to the ideal state. In the higher-quality meshes, a skewness average of 0.21 and an orthogonality quality average of 0.78 generated a high-quality mesh.

### 4.2. Mass, Momentum, Energy, and Turbulence Equations

Among the commonly used turbulence models, the tunnel airflow temperature calculated by the k-ε model is closest to the actual situation and provides the best results for the simulation of airflow temperature [38,39]. In the roadway’s flow, turbulent mass, momentum, and energy transport coincide. The mass, momentum, and energy conservation equations can be expressed as follows:

Continuity equation:(2)$$\frac{\partial \rho }{\partial t}+\nabla \cdot \left(\rho U\right)=0$$

Momentum equation:(3)$$\frac{\partial }{\partial t}\left(\rho U\right)+\nabla \cdot \left(\rho UU\right)=-\nabla P+\nabla \cdot \left(\overset{=}{\tau }\right)+\rho \vec{g}$$
where ρ is air density (kg/m^3^), *P* is the static pressure (Pa), *U* is the velocity vector (m/s), $\overset{=}{\tau }$ is the stress tensor (Pa), and $\rho \vec{g}$ is the gravitational body force.

Conservation of energy is described as follows:(4)$$\frac{\partial }{\partial t}\left(\rho E\right)+\nabla \cdot \left(U\left(\rho E+\rho \right)\right)=-\nabla \cdot \left(\underset{j}{\sum }h_{j}J_{j}\right)+S_{q}$$
where *E* is total energy (J), *h_j_* is the formation enthalpy of species *j* (J/kg), *J_j_* is the diffusion flux of species *j* (kg/(m^2^∙s)), *S_q_* includes the heat of the chemical reaction and any other volumetric heat sources (*J*).

The turbulence kinetic energy, *k*, and its rate of dissipation, *ε*, were obtained from the following transport equations:(5)$$\frac{\partial }{\partial t}\left(\rho k\right)+\nabla \cdot \left(\rho Uk\right)=\nabla \cdot \left[\left(\mu +\frac{\mu_{t}}{\sigma_{k}}\right)\nabla k\right]+G_{k}-\rho \varepsilon$$
(6)$$\frac{\partial }{\partial t}\left(\rho \varepsilon \right)+\nabla \cdot \left(\rho U\varepsilon \right)=\nabla \cdot \left[\left(\mu +\frac{\mu_{t}}{\sigma_{\varepsilon }}\right)\nabla \varepsilon \right]+C_{1\varepsilon }\frac{\varepsilon G_{k}}{k}-C_{2\varepsilon }\rho \frac{\varepsilon^{2}}{k}$$
(7)$$\mu_{t}=\rho C_{\mu }\frac{k^{2}}{\varepsilon }$$
where *G_k_* represents the generation of turbulence kinetic energy due to the mean velocity gradients, *μ* the dynamic viscosity of the fluid (Pa∙s), *C_1ε_* and *C_2ε_* are model constants, *σ_k_* and *σ_ε_* are the turbulent Prandtl numbers corresponding to the *k* equation and the *ε* equation, respectively, and *μ_t_* is turbulent viscosity (Pa∙s).

### 4.3. Assumptions and Boundary Conditions

The main assumptions commonly used for coupled rock–airflow heat-ransfer models are the following [40]:(1)The incompressible ideal gas law was applied to define the fluid density.(2)The heat loss caused by the work done by the viscous force of the fluid was ignored.(3)The airflow was fully developed and turbulent.(4)The turbulent viscosity was isotropic, and the turbulent viscosity coefficient could be considered a scalar.(5)The thermal continuity involving stationary and no-slip conditions was applied to the outer surface of the heading and between the inner surface of the airway and the air.(6)The rock surrounding the roadway was isotropic, homogeneous, and fissure-free.(7)The wall of the roadway and its internal surrounding rock were dry.(8)There was no internal heat source in the alleyway.(9)The constant temperature, regarded as the virgin rock temperature, was applied to the far-field boundary condition.

According to the actual coal mine site, the heat source in the roadway was mainly the surrounding rock. Heat sources such as ventilation equipment, pumps, lighting facilities, and operators generated little scattered heat, so they were neglected. The velocity of the airflow inlet was 2.5 m/s, and the temperature was 296.15 K; the outlet used a pressure outlet. According to the geological data for the coal mine, the temperature of the surrounding rock remained constant beyond 30 m. The rock temperature at 30 m was 323.15 K. Therefore, the fixed temperature (323.15 K) boundary condition was applied to the far-field boundary of the surrounding rock. The coupled boundary conditions were applied by creating cross interfaces between the surrounding rock and the outside of the support layer and between the roadway surface and the air, so that fluid heat transfer and solid heat transfer were coupled. The primary rock type of the surrounding rock was siltstone. An 0.2 m thick concrete support layer was sprayed on the roadway surface to reinforce the roadway. These materials were considered to be isotropic and homogeneous. The settings of the boundary condition parameters in the simulation are shown in Table 7; the thermophysical parameters of the surrounding rock, plain concrete, and insulating concrete are shown in Table 8. The insulating concrete used was concrete with a 15% fiber content. The wall surface of the roadway significantly influenced the turbulent flow and, consequently, the heat exchange. This study considered the effect of roughness by correcting the wall roughness law. Two roughness parameters were specified: roughness height, Ks, and roughness constant, Cs [41].

### 4.4. Simulation Results and Discussion

#### 4.4.1. Wind-Flow and Surrounding-Rock Temperature Field

Figure 14 shows the central longitudinal section of the wind-flow temperature field. In the two temperature fields, we observed two different regions: the heat exchange layer near the roadway wall and the flow layer near the center of the roadway. The temperature rose continuously in the heat transfer layer with the wind flow along the *X*-axis. The closer to the wall, the faster the temperature rose, and the temperature rose significantly in the radial direction of the roadway. In the flow layer, the wind flow temperature was almost unchanged and remained around the inlet temperature (288.65 K). The surrounding-rock temperature was higher than the wind-flow temperature. The wind flow was constantly washing the wall of the roadway. The heat exchange was via convection heat transfer between solid and fluid, with high heat exchange efficiency. However, the heat transfer between the wind streams was thermal convection with low heat-transfer efficiency, resulting in a slight temperature gradient in the flow layer and insignificant heat transfer in the axial direction of the roadway.

Figure 15 shows the cross-section of the temperature field of the surrounding rock. In order to better observe the effect of wind flow on the temperature field of the surrounding rock, a cross-section was set every 20 m from the entrance to the exit of the roadway. We observed that the temperature of the surrounding rock gradually decreased from the far field boundary to the roadway surface, and the closer the distance to the roadway wall, the more significant the decrease in temperature. There was a temperature difference between the surrounding rock and the wind flow. A large amount of heat within the surrounding rock entered the roadway through the supporting concrete, resulting in a continuous decrease in the temperature of the surrounding rock. Compared with ordinary concrete, the temperature field of surrounding rock under-insulated concrete support was less disturbed in scope and degree, and the temperature was higher at the same depth. This was because the insulated concrete had a smaller thermal conductivity, which had a more significant obstruction effect on the heat flow, reducing the efficiency of heat dissipation from the depth of the surrounding rock to the roadway, slowing down the rate of temperature increase in the wind flow, and improving the utilization rate of the cold volume in the wind flow.

Figure 15 also shows that the cooling range of the wind flow to the surrounding rock at the entrance of the roadway was more significant than that at the exit, under both types of concrete support. This was because, as the wind flow flowed, the heat exchange between the wind flow and the wall of the roadway continuously occurred, the temperature of the wind flow kept increasing, the temperature difference between the wind flow and the surrounding rock kept decreasing, and the temperature regulation ability of the wind flow to the surrounding rock then decreased. Its influence range became smaller and smaller. Zeng et al. [42] reached the same conclusion when simulating the temperature field in the Nanshan tunnel, where the tunnel entrance airflow and the surrounding rock temperature changed rapidly. The insulation layer effectively reduced the heat exchange between the airflow and the surrounding rock and reduced the cooling rate of the surrounding rock.

In general, the thermal conductivity of insulated concrete is much smaller than that of ordinary concrete. The wind flow is less affected by the heat dissipation of the surrounding rock, which can make more cold volume transfer to the mining face and play a cooling role.

#### 4.4.2. Thermal-Coupling Interface

When there was a temperature difference between the roadway wind flow and the heat source deep in the surrounding rock, ignoring the heat transfer between the gas and liquid phases inside the rock, the heat transfer process was as shown in Figure 16: heat reached the surrounding rock-concrete intersection through heat conduction inside the rock and was transferred to the concrete through solid-solid heat coupling, and heat again reached the intersection between the roadway wall and the wind flow after heat conduction in the concrete, and was finally transferred to the wind flow through solid-fluid heat coupling. The heat passed through two key coupling interfaces in heat transfer. It was necessary to analyze the temperature distribution of the two cross interfaces and the distribution of the convective-heat-transfer coefficient between the wind flow and the wall, because they affect the total heat dissipation from the surrounding rock.

Figure 17 shows the temperature distribution of the roadway wall under different concrete supports. From the entrance to the roadway exit, the temperature was gradually increasing, which may have been due to the strong cooling capacity of fresh air. The wall temperature of under-plain concrete support ranged from 291.6 K to 299.1 K, with an average temperature of 296.4 K. The wall temperature of under-insulated concrete support ranged from 292.6 K to 296.3 K, with an average temperature of 294.5 K. The wall temperature of the roadway with plain concrete was higher, with an average temperature of 1.9 K higher.

Figure 18 shows the temperature distribution of the wall surface of the surrounding rock under different concrete supports. From the entrance to the roadway exit, the temperature was gradually increasing. The wall surface under-plain concrete support ranged from 296.5 K to 300.2 K, with an average temperature of 298.3 K. The wall surface under-insulated concrete support ranged from 311.9 K to 313.3 K, with an average temperature of 312.6 K.

Under plain-concrete support, the average temperature difference between the two walls was 1.9 K, while with under-insulated concrete support, the average temperature difference between the two walls was 15.1 K. This showed that insulated concrete can effectively keep heat out of the roadway.

Figure 19 shows the distribution of the convective heat transfer coefficient with the inlet temperature as the reference temperature. From the inlet to the outlet of the roadway, the convective heat transfer coefficient increased rapidly and then stabilized before slowly decreasing. This may have been due to the intense heat exchange between the fresh cold air and the wall. However, as the temperature difference between the wind flow and the surrounding rock decreased along the *X*-axis, the heat transfer between the surrounding rock and the airflow was gradually reduced. The average value of the convective-heat-transfer coefficient was 7.02 W/(m^2^·°C), and the maximum value was 7.47 W/(m^2^·°C) in the area of intense heat transfer under plain-concrete support, which was approximately 30 m. The area of intense heat transfer under the support of insulating concrete, approximately 25 m, had an average convective heat transfer coefficient of 6.01 W/(m^2^·°C) and a maximum value of 6.42 W/(m^2^·°C). The convective-heat-transfer intensity between ordinary concrete with larger thermal conductivity and wind flow was more significant, and more heat was emitted from the surrounding rock.

#### 4.4.3. Airflow Temperature Field at the Outlet

The airflow flows out of the section of the concentrated flat lane and enters the air cooler of the mining work surface. If the airflow temperature is low, the cooling power of the air cooler can be reduced, saving energy. Figure 20 shows the temperature field of the exit airflow from the section-concentrated flat lane. We observed that in both cases, there iwass an unheated circular area (flow layer) at the center of the roadway section where the temperature was kept at 288.65 K, and the other areas had higher temperatures closer to the wall. The radius of the flow layer under plain-concrete support was approximately 1.46 m, the average value of wind flow temperature was 289.29 K, and the maximum value was 290.10 K. The radius of the flow layer under-insulated concrete support was approximately 2.10 m, the average value of wind-flow temperature was 288.99 K, and the maximum value was 289.34 K. The unheated area under-insulated concrete support was much larger, and the average temperature of the exit airflow increased by 0.34 K, compared with the temperature of the inlet airflow (the average temperature of the exit of the roadway under plain-concrete support increased by 0.64 K). The sprayed insulating concrete on the roadway’s surface effectively improved the utilization of the cold volume in the wind flow. It achieved the purpose of reducing the temperature of the wind flow.

## 5. Cost-Benefit Analysis

The raw-rock temperature of the mine studied in this paper was more than 45 °C. The heat damage was severe, especially in summer, and mechanical cooling was required throughout the year. We believe that cost-benefit analysis plays an essential role in cooling design and construction when selecting cooling methods and determining the cost and benefit of the entire cooling project. Therefore, the integrated costs of jetted insulated concrete must be analyzed.
(8)$$M_{z}=M_{1}+M_{2}+M_{3}$$
where $M_{z}$ is the comprehensive cost, $M_{1}$ represents the material costs, $M_{2}$ represents the construction and management costs, and $M_{3}$ represents the maintenance management costs.

The formula shows that the integrated costs include material costs, construction, and management costs, and maintenance management costs [43]. Compared with plain concrete, the changes in construction procedures and technical requirements are minimal, and the construction and management costs will not increase significantly. In contrast, the compressive strength of insulating concrete is less than plain concrete, and the later maintenance and management costs may increase; therefore, the increased costs are mainly focused on material costs and maintenance management costs.

The ratio used for thermal-insulating concrete was ceramsite and glazed-hollow-bead composite concrete material with 15% basalt fiber, and the spraying thickness was 0.2 m. According to the market survey, the raw materials and costs needed per square meter of thermal insulating concrete and plain concrete are shown in Table 9 and Table 10.

The material cost per square meter of insulating concrete is USD 4.584, while the material cost per square meter of plain concrete is USD 1.104. The increase in cost per square meter of the alleyway is approximately USD 3.57.

The Newtonian cooling equation is often used to determine the heat load of development [44].
(9)$$Q=\alpha \left(t_{w}-t_{f}\right)A$$
where Q is the heat flux (w), α is the convective heat-transfer coefficient between the surrounding strata and the airflow (W/(m^2^·°C)), $t_{w}$ is the wall temperature of the surrounding strata (k), $t_{f}$ is the airflow temperature (k), and A is the area of the heat-transfer region (m^2^).

In this study, Newton’s cooling equation was used to calculate the difference value in heat dissipation between the surrounding rock under plain concrete and insulated concrete support; then, the reduced electricity cost of the mechanical cooling system under-insulated concrete support was calculated.
(10)$$\nabla Q=Q_{p}-Q_{g}$$
where ∇Q is the difference in heat release from surrounding rock (w), $Q_{p}$ is the total amount of heat dissipation from the surrounding rock under ordinary concrete support (w), and $Q_{g}$ is the total heat dissipation from the surrounding rock supported by insulating concrete (w).

The annual power-cost equation for the mine mechanical-cooling system is the following.
(11)$$M_{j}=\nabla Q\times 24\times 365\times e\times c^{-1}\times d^{-1}\times 10^{-3}$$
where $M_{j}$ is the the cost of electricity consumption for one year ($), e is the price of electricity for industrial weekdays (kW·h/$), c is the cooling efficiency of mechanical refrigeration cooling systems (%), and d is the motor efficiency of compressors for mechanical refrigeration cooling systems (%).

This simulated section of the centralized flat lane has a long service life, and the cost-effectiveness of spraying insulated concrete on the roadway’s surface was considered for 5 years. The average temperature of the wall of the plain concrete roadway was simulated to be 296.4 K, and the convective-heat-transfer coefficient was 7.02 W/(m^2^∙°C). The average temperature of the wall of the insulated concrete roadway was 294.5 K, and the convective-heat-transfer coefficient was 6.01 W/(m^2^∙°C). The wind-flow temperature was 288.62 K. The electricity price in the industrial sector was 0.105 kW∙h/$; the refrigeration efficiency of the mechanical refrigeration cooling system was 85%; and the motor efficiency of the compressor was 89%. The cost of mechanical cooling for 5 years was USD 329.54 per square meter of the roadway under plain concrete support, and USD 212.96 per square meter of under-insulated concrete support. According to the coal mine information and the related technical personnel, the maintenance and management cost per square meter of insulating concrete will increase by about USD 30.5 in 5 years, so the cost saving by spraying insulating concrete on the roadway wall can be USD 82.51 in 5 years, which is an excellent economic measure and suitable for promotion.

## 6. Conclusions

The feasibility and cost-effectiveness of insulating concrete with the prepared basalt fiber to achieve a high geothermal mine-cooling design were analyzed by performance tests and numerical simulations. The following conclusions can be drawn.

(1) The physical property indices of basalt -iber-insulated concrete were derived by testing the apparent density, compressive strength, permeability, and thermal conductivity. The apparent density of the dried specimens containing 20% fiber was reduced by about 18%, compared with the reference specimens. Evaluating the compressive strength of fiber-insulated concrete highlighted the detrimental effects of high-fiber content. The compressive strengths were lower than those of the reference specimen (0%), from 36.7 MPa to 17.4 MPa. However, the compressive strengths of the fiber concrete for all fiber contents, except for the 20% fiber content, met the standard. The appropriate amount of fiber content can improve the impermeability of concrete to a certain extent. The impermeability of concrete with 5% and 10% fiber content increased by 22.5% and 4.2%, respectively. The thermal conductivity of insulating concrete containing 20% fiber in the wet state was reduced to 0.098 W/(m∙°C), and the thermal insulation performance was increased by approximately 74.5%. Although the thermal conductivity of insulated concrete showed an increasing trend with increasing temperature, the increase was minimal. Finally, the optimum fiber content in concrete was analyzed by the correlation matrix method, with the 15% fiber content showing the best overall performance.

(2) The thermal insulation performance and insulation mechanism of basalt-fiber concrete were investigated by numerical simulation. The wall temperatures and convective heat transfer coefficients were much lower in shotcrete-insulated concrete roadways, compared with plain concrete roadways, while the surrounding rock temperatures were higher. The insulation of concrete can effectively slow down the heat dissipation rate from the surrounding rock and improve the utilization of the cold volume in the wind flow. Compared with plain concrete, the average airflow temperature at the exit of a 100 m long tunnel supported by insulated concrete decreased by 0.3 K.

(3) The reduction of heat dissipation per square meter of surrounding rock after spraying basalt-fiber-insulated concrete, and the electricity cost savings, were calculated by Newton’s law of cooling, which verified the good economic benefits of insulated concrete.

Basalt-fiber-insulated concrete is a promising application. Further studies on the size, orientation, and distribution of basalt fibers in concrete are needed to further improve the thermal and mechanical properties of concrete.

## Figures and Tables

**Figure 1 materials-15-08236-f001:**
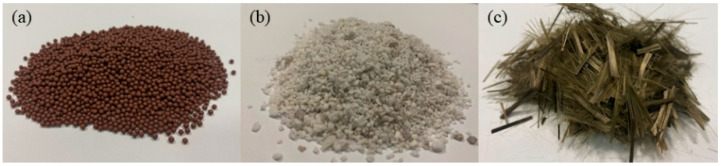
(**a**) Ceramsite; (**b**) glazed hollow beads; (**c**) basalt fiber.

**Figure 2 materials-15-08236-f002:**
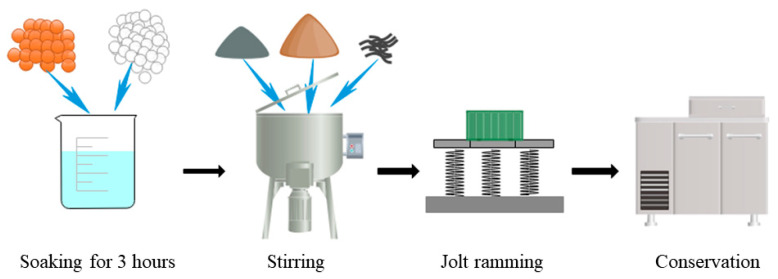
Specimen production process.

**Figure 3 materials-15-08236-f003:**
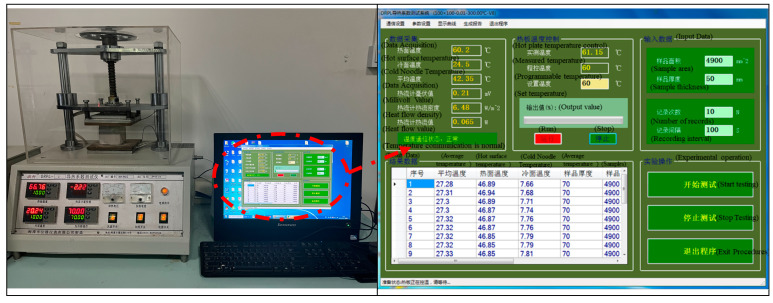
Measurement of the thermal conductivity of the specimen. Left subfigure: DRPL-I thermal conductivity tester; right subfigure: DRPL test system.

**Figure 4 materials-15-08236-f004:**
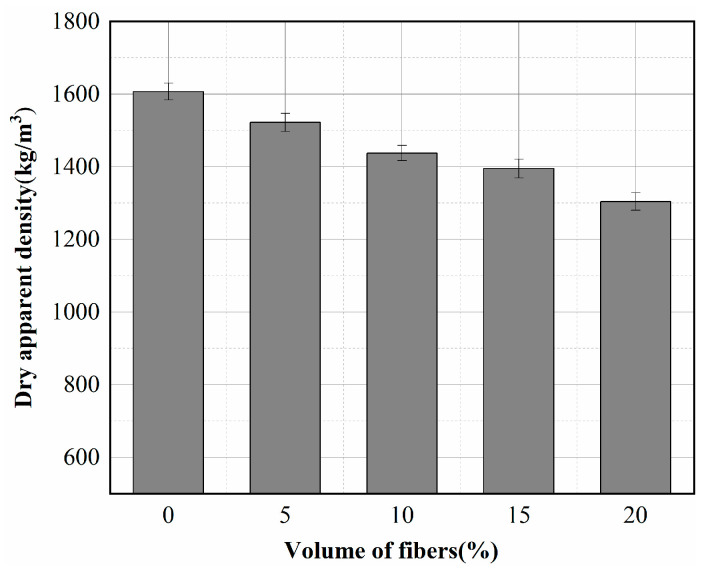
Apparent density of specimens with different fiber contents.

**Figure 5 materials-15-08236-f005:**
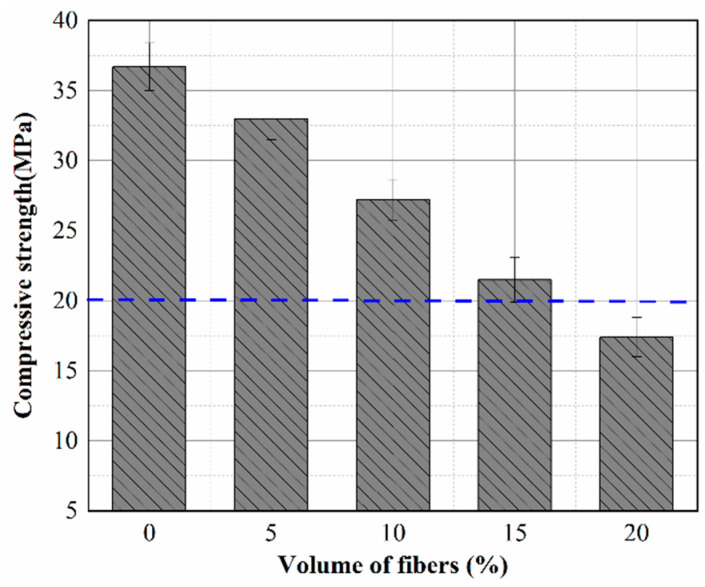
Effect of fiber content on the strength of thermal-insulating concrete.

**Figure 6 materials-15-08236-f006:**
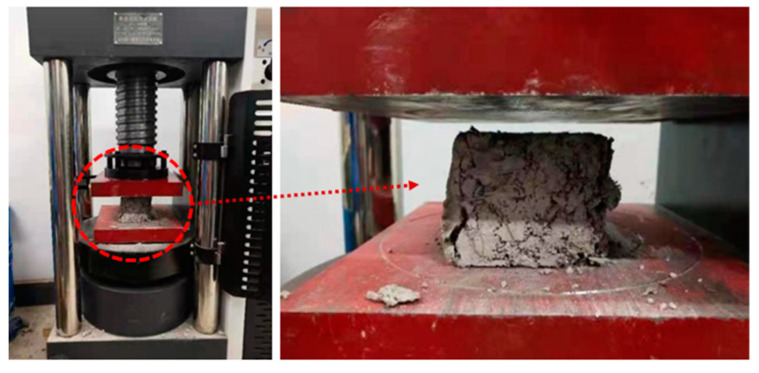
Damage diagram of 20% fiber concrete specimen.

**Figure 7 materials-15-08236-f007:**
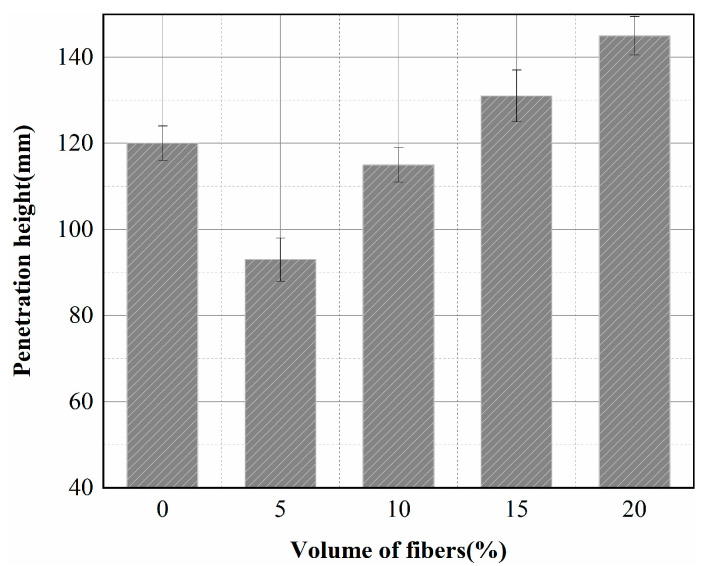
Effect of fiber content on the impermeability of thermal insulation concrete.

**Figure 8 materials-15-08236-f008:**
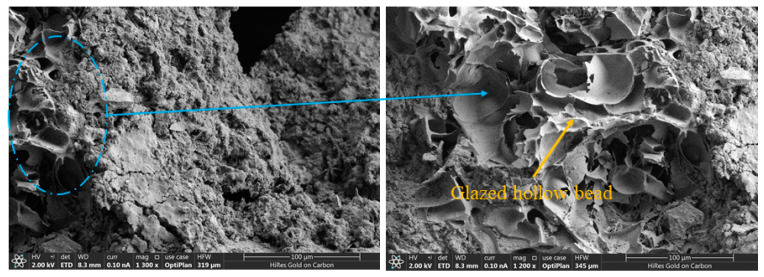
SEM image of the reference specimen.

**Figure 9 materials-15-08236-f009:**
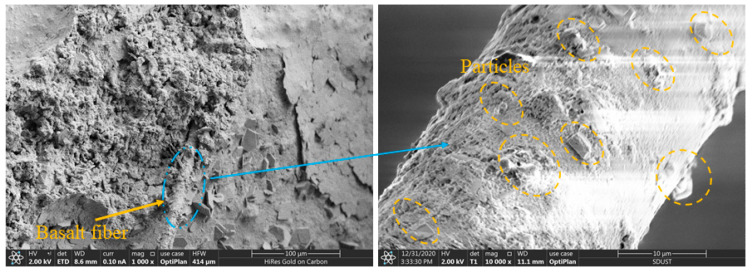
SEM image of thermal-insulation concrete with 5% fiber.

**Figure 10 materials-15-08236-f010:**
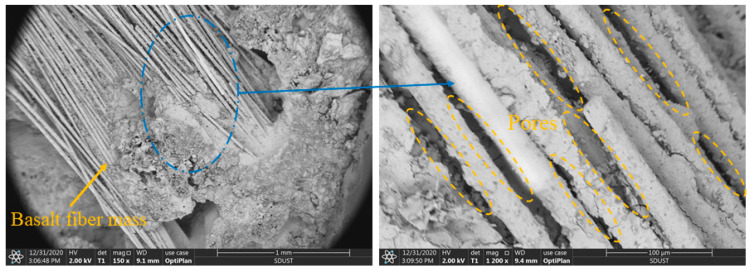
SEM image of thermal-insulation concrete with 20% fiber.

**Figure 11 materials-15-08236-f011:**
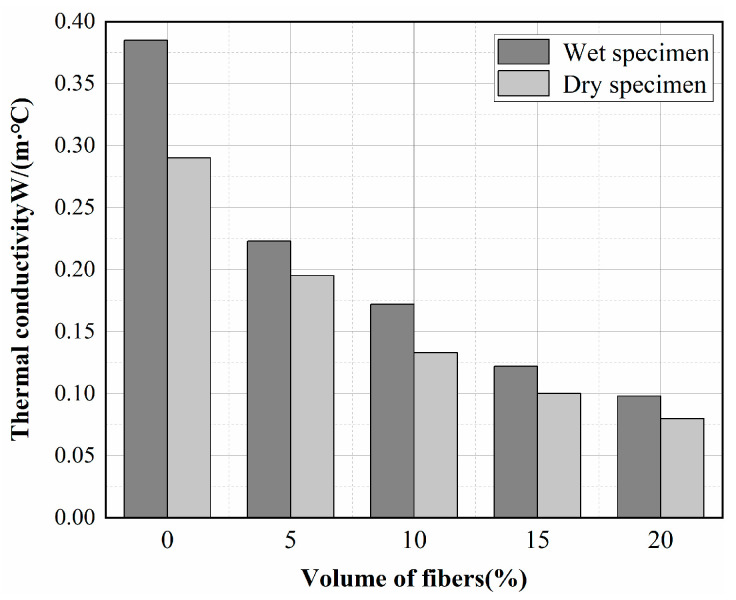
Effect of fiber content on the thermal conductivity of insulating concrete.

**Figure 12 materials-15-08236-f012:**
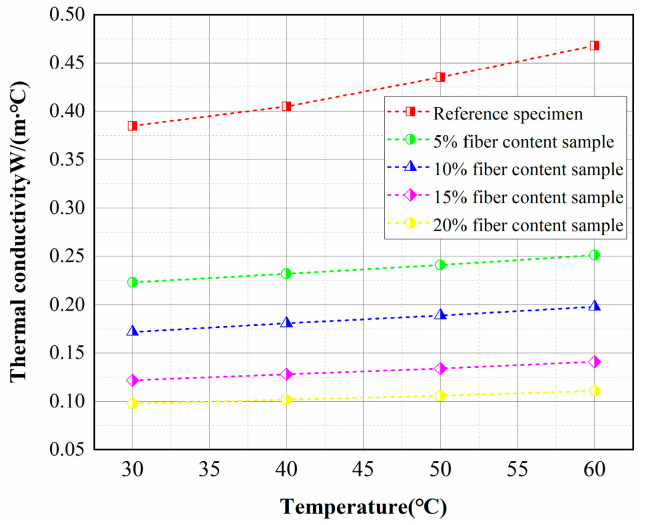
Effect of temperature on the thermal conductivity of insulating concrete.

**Figure 13 materials-15-08236-f013:**
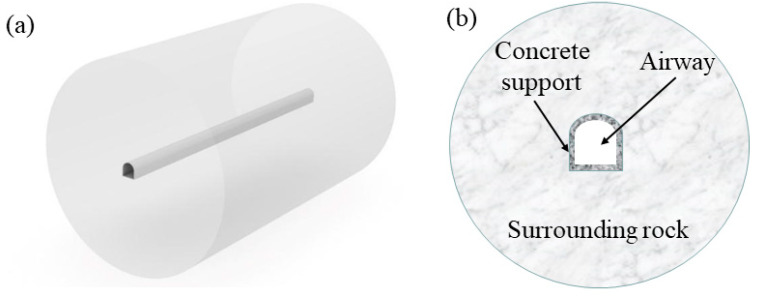
Three-dimensional geometric model: (**a**) stereo view; (**b**) cross-section.

**Figure 14 materials-15-08236-f014:**
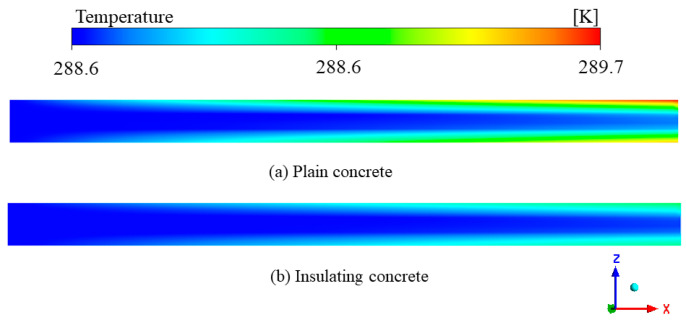
Longitudinal section of wind-flow temperature field: (**a**) Wind flow temperature field of the roadway under normal concrete support; (**b**) Wind flow temperature field of the roadway under insulated concrete support.

**Figure 15 materials-15-08236-f015:**
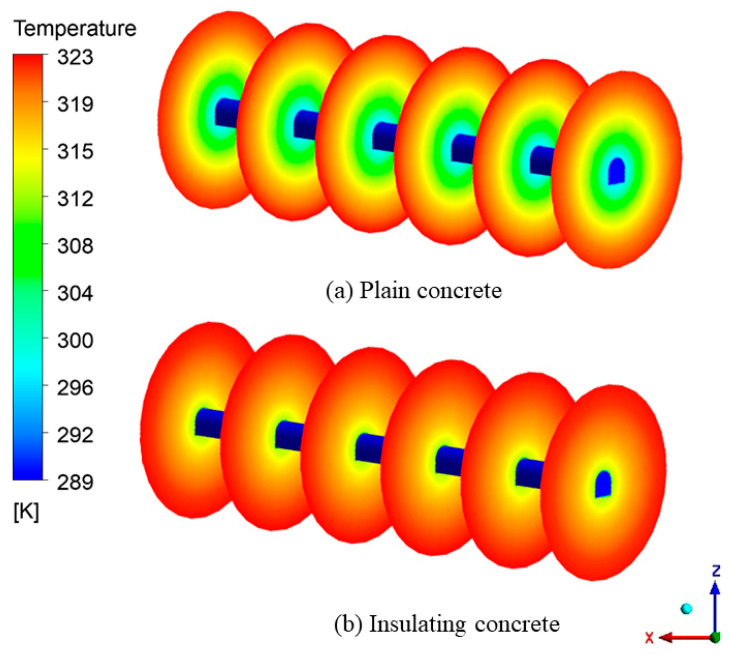
Surrounding-rock temperature field: (**a**) Temperature field of the surrounding rock of the roadway under plain concrete support; (**b**) Temperature field of the surrounding rock of the roadway under insulated concrete support.

**Figure 16 materials-15-08236-f016:**
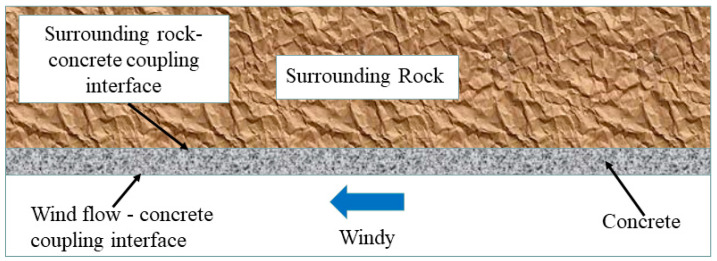
Heat transfer schematic.

**Figure 17 materials-15-08236-f017:**
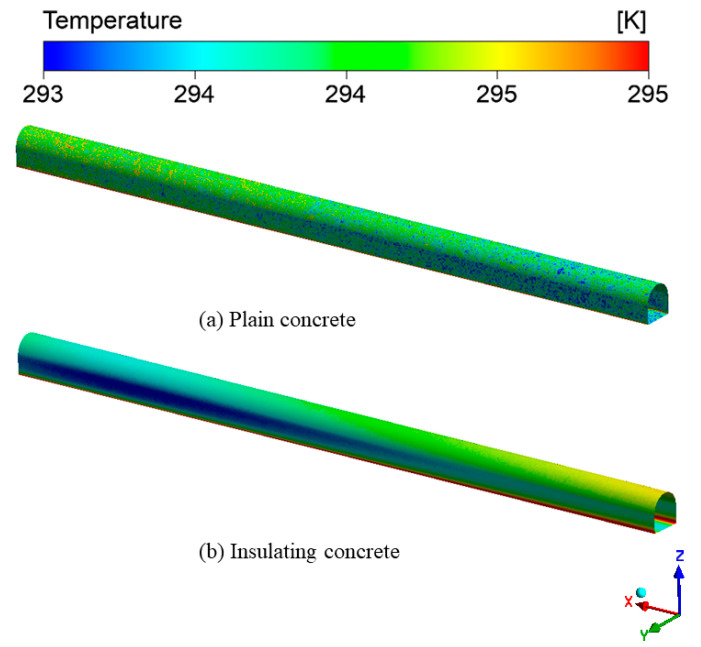
Temperature distribution on the wall of the roadway: (**a**) Temperature distribution on the wall of the roadway under plain concrete support; (**b**) Temperature distribution on the wall of the roadway under insulated concrete support.

**Figure 18 materials-15-08236-f018:**
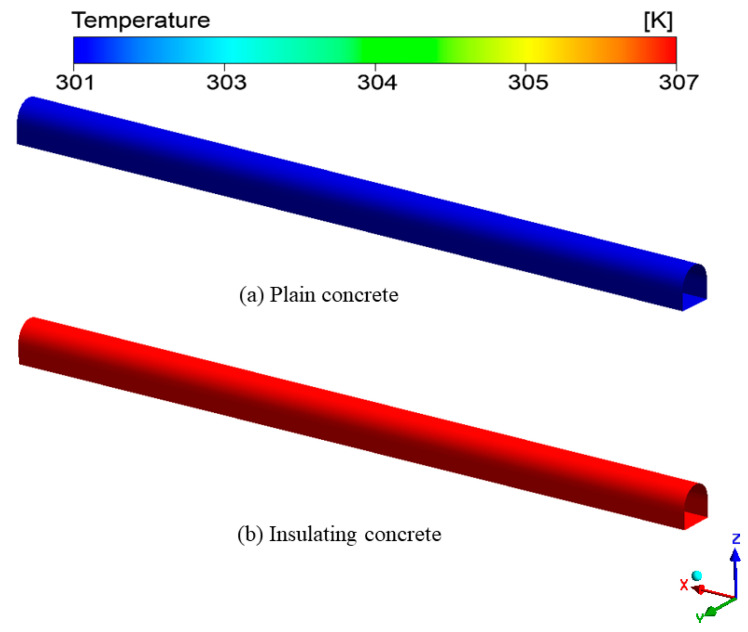
Temperature distribution of the wall surface of the surrounding rock: (**a**) Temperature distribution on the wall surface of the surrounding rock in the roadway under plain concrete support; (**b**) Temperature distribution on the wall surface of the surrounding rock in the roadway under insulated concrete support.

**Figure 19 materials-15-08236-f019:**
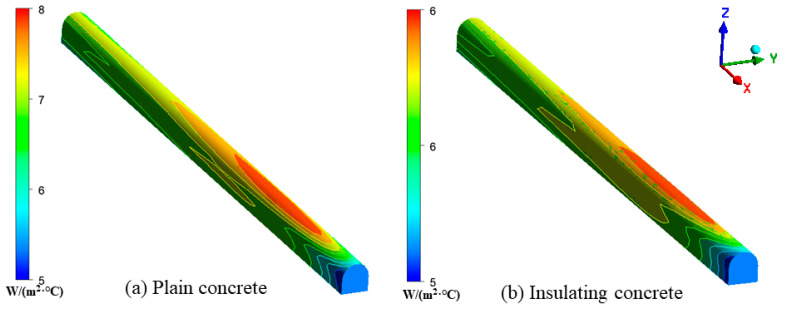
Distribution of convective-heat-transfer coefficients between the wall of the roadway and the wind flow: (**a**) Distribution of convective heat transfer coefficients of the roadway under plain concrete support; (**b**) Distribution of convective heat transfer coefficients of the roadway under insulated concrete support.

**Figure 20 materials-15-08236-f020:**
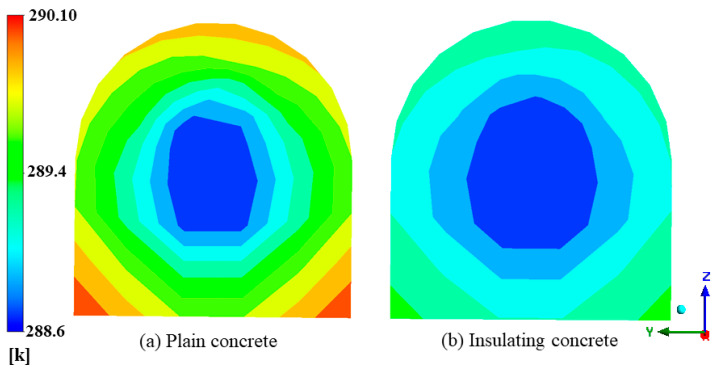
Wind-flow temperature field at the exit of the roadway: (**a**) Wind flow temperature field at the exit of the roadway under plain concrete support; (**b**) Wind flow temperature field at the exit of the roadway under insulated concrete support.

**Table 1 materials-15-08236-t001:** Physical properties of ceramsite.

Particle Size (mm)	Loose Bulk Density (kg/m^3^)	Numerical Tube Pressure (MPa)	Water Absorption (%)	Thermal Conductivity ((W/(m·°C))	Porosity (%)
~0.8–1.5	600	3.5	7	0.039	25

**Table 2 materials-15-08236-t002:** Physical properties of glazed hollow beads.

Particle Size (mm)	Bulk Density (kg/m^3^)	Thermal Conductivity (W/(m·°C))	Closed Cell Rate (%)	Water Absorption (%)
~0.5–1.0	80	0.027	96	8

**Table 3 materials-15-08236-t003:** Physical properties of basalt fiber.

Length (mm)	Diameter (μm)	Density (g/cm^3^)	Tensile Strength (MPa)	Ultimate Elongation (%)
10	15	2.65	~3300–4500	~2.4–3.0

**Table 4 materials-15-08236-t004:** Fiber content evaluation results.

Fiber Content	Apparent Density	Compressive Strength	Impermeability	Thermal Insulation
5%	heavier	excellent	excellent	general
10%	lighter	very good	very good	good
15%	light	good	good	very good
20%	very light	general	general	excellent

**Table 5 materials-15-08236-t005:** Weighting of evaluation indicators.

Indicators	Pair-by-Pair Comparison						Score	Weight Function
	1	2	3	4	5	6		
Apparent density	0	0	0				0	0
Compressive strength	1			1	0		2	1/3
Impermeability		1		0		0	1	1/6
Thermal insulation			1		1	1	3	1/2

**Table 6 materials-15-08236-t006:** Systematic evaluation correlation matrix.

Fiber Content	Apparent Density	Compressive Strength	Impermeability	Thermal Insulation	*V_i_*
	0	1/3	1/6	1/2	
5%	1	5	4	3	3
10%	2	4	3	4	3.8
15%	3	3	2	5	3.83
20%	4	2	1	6	3.67

**Table 7 materials-15-08236-t007:** Setting of simulation parameters.

Type	Property	Value	Type	Property	Value
Inlet	Velocity inlet	2.5 m/s	Wall of airway	Ks	0.02 m
	Temperature	15.5 °C		Cs	0.05
Outlet	Pressure outlet	1.1 atm	Solution methods	Scheme	Coupled
Viscous model	K-epsilon	Standard	Spatial discretization	Turbulent kinetic energy	Second-order upwind
Far-Field boundary	Temperature	50 °C		Turbulent dissipation rate	Second-order upwind
General	Solver type	Pressure-based		Pressure	PRESTO!

**Table 8 materials-15-08236-t008:** Thermophysical parameters of the surrounding rock, the support, and the air.

Material	Thermal Conductivity (W/(m∙°C))	Density (kg/m^3^)	Specific Heat (J/(kg∙°C))
surrounding rock	5.1	2593	790
plain concrete	3.4	2400	1000
insulating concrete	0.134	1650	1120
air	0.0242	1.225	1006.43

**Table 9 materials-15-08236-t009:** Raw materials and costs per square meter of insulating concrete.

Name	Cement	Sand	Glazed Hollow Bead	Ceramisite	Water	Basalt Fiber
Weight (g)	6997	113,364	696	7136	5438	2320
Unit price ($/t)	64.94	8.66	216.45	79.37	0.60	1414.14
Cost ($)	0.46	0.12	0.15	0.57	0.004	3.28

**Table 10 materials-15-08236-t010:** Raw materials and costs per square meter of plain concrete.

Name	Cement	Sand	Rocks	Water
Weight (g)	10,440	20,880	20,880	5220
Unit price ($/t)	64.94	8.66	7.22	0.60
Cost ($)	0.68	0.18	0.15	0.004

## Data Availability

The study did not report any data.

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
