# Peer review of "Performance Test and Thermal Insulation Effect Analysis of Basalt-Fiber Concrete"

_materials, 2022, doi:10.3390/ma15228236_

Round 1
Reviewer 1 Report
The authors had made a significant contribution by studying s the feasibility of applying inorganic thermal insulating concrete in high-ground temperature roadways in underground coal mines. Following are my suggestions.
1. Please modify the title and make it more specific.
2. Add more relevant keywords.
3. Please improve the introduction. Add more about benefits of basalt fibers and types of FEM in the introduction. Please refer to https://doi.org/10.1016/j.jobe.2020.101689 ;https://doi.org/10.1002/fam.2968 ; https://doi.org/10.1016/j.conbuildmat.2022.126340
4. Please write clearly the objective and significance of your work in the last paragraph of introduction.
5. Please improve the Figure 2 and 3 quality.
6. A comparison with previous study is necessary.
7. Add a discussion section before conclusion regarding practical implementation of current study.
8. Please make bullet point in conclusion.
9. What are your future recommendation.
10. Moderate English changes required.
11. References are too old. Please update.
12. Overall the study is good and results are impressive.
Reviewer 2 Report
1- The Abstract is 394 words; I suggest the authors reduce it to approximately 250-270 words.
2- The authors should present a table for all information between lines 134-141.
3- The authors should use a bigger size for figure 3; the information and images are not clear.
4- I suggest, For input parameters (mixed design), the authors use a correlation matrix graph.
5- The authors should compare their results with standard codes and other researchers. It is essential for the validation of their results.
Round 2
Reviewer 2 Report
The paper improved, and the authors responded to all comments.